# Cost-effectiveness of sentinel screening of endemic diseases alongside malaria diagnosis: A case study in schistosomiasis

**Francesco Manca**[1]*, **Giorgio Ciminata**[1], **Eleanor Grieve**[1], **Julien Reboud**[2], **Jonathan Cooper**[2], **Emma McIntosh**[1]

**1** School of health and Wellbeing, University of Glasgow, Glasgow, United Kingdom, **2** James Watt School of Engineering, University of Glasgow, Glasgow, United Kingdom

* francesco.manca@glasgow.ac.uk

## Abstract

### Background

In countries where malaria is endemic, the use of rapid diagnostic tests(RDTs) has become routine, especially in rural settings. Such regions are characterised by often having other co-endemic infectious diseases, at high levels of prevalence.

### Aim

To illustrate the potential added-value of "sentinel" screening for patients presenting for a routine diagnostic test for malaria, at healthcare facilities in Uganda.

### Methods

We developed an economic model by combining two decision trees, one for malaria and a second for the co-endemic disease schistosomiasis. The integrated model was designed to inform policy strategies for the co-endemic disease in addition to malaria (i.e., whether to test opportunistically for schistosomiasis or use mass drug administration(MDA) as per usual practice).We performed the analysis on three comparators varying testing accuracy and costs.

### Results

Sentinel screening can provide added value to the testing of patients compared with the status quo: when schistosomiasis prevalence is high then MDA is preferential; if low prevalence, treating no one is preferred. If the disease has average levels of prevalence, then a strategy involving testing is preferred. Prevalence thresholds driving the dominant strategy are dependent upon the model parameters, which are highly context specific. At average levels of prevalence for schistosomiasis and malaria for Uganda, adding a sentinel screening was cost-effective when the accuracy of test was higher than current diagnostics and when economies of scope were generated(Expected value clinical Information = 0.65$ per

**Data Availability Statement:** All relevant data are within the manuscript and its Supplementary Material.

**Funding:** The study was supported by the Engineering and Physical Sciences Research Council (EPSRC) Institutional Support Fund (grant no. EP/R512813/1, JC), as well as by EPSRC EP/R01437X/1 (JC, JR, EM, GC), European Union Horizon Europe (101057251, JC, JR), UKRI Innovate UK (10052860, JC, JR), the National Institute for Health Research and EP/T029765/1 (JC, JR) and Royal Academy of Engineering Research Chair in Global Health (RCSRF2223-15-51, JC). The funders had no role in study design, data collection and analysis, decision to publish, or preparation of the manuscript.

**Competing interests:** The authors have declared that no competing interests exist.

DALY averted, 137.91$ per correct diagnoses).Protocols using diagnostics with current accuracy levels were preferred only for levels of MDA coverage below 75%.

## Conclusion

The importance of the epidemiological setting is crucial in determining the best cost-effective strategy for detecting endemic disease. Economies of scope can make sentinel screenings cost-effective strategies in specific contexts. Blanket thresholds recommended for MDA may not always be the preferred option for endemic diseases.

## Author summary

Malaria tests are commonly the default assessment when febrile patients arrive at clinics in low- and middle-income countries. This is due to the high prevalence of the disease coupled with the increased reliability and affordability of rapid diagnostic tests. While malaria and other prevalent diseases have received great attention in terms of international investments and local efforts over the years, other endemic diseases do not receive the same consideration in terms of testing capacity and, consequently, correct treatment. This study aims to provide a model to inform policymakers on the costs and benefits of carrying out an opportunistic screening for one (or potentially more) of these neglected tropical diseases (NTD), performed at the same time as a malaria diagnosis for febrile patients presenting at clinics. To achieve this, we developed a model to evaluate different healthcare protocols detecting malaria and schistosomiasis simultaneously, with input values referring to Uganda, as a case study. Schistosomiasis is an NTD and the main treatment strategy in endemic populations is mass drug administration (MDA). This study shows the added value of sentinel screening during a malaria diagnosis varying the level of prevalence of both malaria and schistosomiasis.

## Introduction

In regions where malaria is endemic, rapid diagnostic tests (RDTs) are now commonly used [1], providing an easy to use diagnostic compared to gold standard microscopy tests [2]. The cost effectiveness of RDTs, when compared with other diagnostics for malaria is, however context-dependent as the total cost of care, including both diagnosis and treatment depends on parasite prevalence and the available facilities [3–5]. Currently, RDTs are more likely to be cost-effective in rural settings [4,6].

In 2018, 412 million RDTs for malaria were sold globally, with most of them supplied in sub-Saharan Africa, where they represented 80% of all malaria diagnoses [7]. In endemic areas, RDTs have become part of the routine clinical diagnosis approach to febrile illness, where laboratory services to support clinical practice are rarely available (so contributing to reduced misdiagnoses and consequent mistreatments). While malaria has received much international investment, research into other endemic diseases, including many neglected tropical diseases (NTD), have not received the same consideration in terms of testing capacity and/or surveillance.

Mass drug administration (MDA), defined as the administration of drugs to the whole population regardless of its positivity to the disease, is currently the preferred intervention for controlling NTDs such as schistosomiasis [8]. New strategies for treating such diseases may have

the potential to be of value, both in terms of economic and health outcomes. Given the World Health Organization's 2030 roadmap for helminth infection control programmes, including schistosomiasis, which targets the elimination of helminth-attributable morbidity in school-age children with an associated reduction in prescriptions needed in MDA programs, it is possible that rethinking how effective NTD diagnostic techniques can be delivered in progress toward these milestones, is an important and worthy target [9].

Adding 'opportunistic' screening, also known as 'sentinel' screening, for lesser known or asymptomatic diseases, such as schistosomiasis, alongside a malaria diagnosis based on RDTs could be a step change in surveillance and diagnoses of such conditions, providing more cost-effective care.

Furthermore, new challenges to both malaria and schistosomiasis diagnoses and interventions have arisen recently. For instance, many RDT tests now have a reduced efficacy due to plasmodium avoidance hrp2/3 deletions [10] (prevalent feature in common RDTs), or new initiatives in reducing preventative chemotherapy using anti-helminth treatments in MDAs, especially where soil-transmitted helminths and schistosomiasis are co-endemic [9,11]. Whilst the cost-effectiveness of malaria RDTs has been studied extensively compared with other diagnostics, the added value of sentinel screening for NTDs alongside malaria RDTs has not yet been fully considered.

In sub-Saharan Africa alone, mortality and morbidity due to malaria account for 43.5 million of years of life lost (YLL) [12], being the third leading cause of death for children under 5 years of age. Schistosomiasis still affects almost 240 million people worldwide with at least 90% of them living in Africa [13]. It can cause anaemia, stunted growth with consequent poor educational outcomes in children, and if untreated, chronic disease affects the ability to work and can lead to death. Schistosomiasis and malaria are often co-endemic, particularly in rural communities, presenting development challenges for many countries in the Global South [14]. Both diseases result in morbidity and mortality and impact on local health inequalities, with populations in low resource rural or urban settings disproportionately affected through reduced access to diagnosis, treatment, and care.

This study reports the development of an economic model that can support decisions on whether to implement a new healthcare protocol consisting of sentinel screening for endemic diseases alongside malaria diagnosis for individuals presenting at facilities.

Although we focus on the two endemic diseases of malaria and schistosomiasis, sensitivity analyses should also point to the direction of the change in cost-effectiveness when context or disease specific assumptions change.

This study aims to provide a flexible and adaptable model of the value of clinical information of performing a sentinel screening during a routine diagnostic test for malaria. We populate the model with a hypothetical healthcare protocol involving sentinel screening for schistosomiasis alongside malaria testing for individuals presenting at facilities using examples of alternative diagnostics. We compare MDAs for schistosomiasis with three scenarios; the first uses existing diagnostics; the second uses the same but with a reduction in price (due to economies of scope generated by the simultaneous test of two different diseases); the final scenario involves the evaluation of a device prototype able to detect simultaneously multiple diseases and with an increase in test accuracy. Such a device prototype and novel point-of-care DNA assay device has been developed in collaboration with University of Glasgow, and the Ministry of Health, Uganda and has been tested in rural communities [15]. This low-cost point-of-care diagnostic is multiplexed and enables the assignment of malaria species upon detection. The multiplex technology is flexible and can also allow the diagnosis of other diseases, such as schistosomiasis, at the same time, and on the same device, thereby increasing the efficiency of healthcare testing in often hard to reach settings.

## Methods

### Model background

To develop our economic model, we followed the approach of Phelps and Mushin [16], who modelled a diagnostic test at different levels of disease prevalence, focusing on the net utility of the patients undertaking a treatment as a consequence of the diagnostic outcome.

The cost effectiveness of the diagnostic device is modelled as a function of its accuracy (sensitivity and specificity), the cost of the device itself and the economic costs and health outcomes relative to the disease itself (disutility, cost of treatment, etc.). Fig 1, adapting Phelps and Mushin' framework, summarises these concepts providing a visual insight on the expected value of clinical information (EVCI) (expressed in incremental net monetary benefit), and on the relevant drivers for its cost effectiveness [16]. In Fig 1, between prevalence A and B, it would be recommended to use the diagnostic device as the incremental net monetary benefit

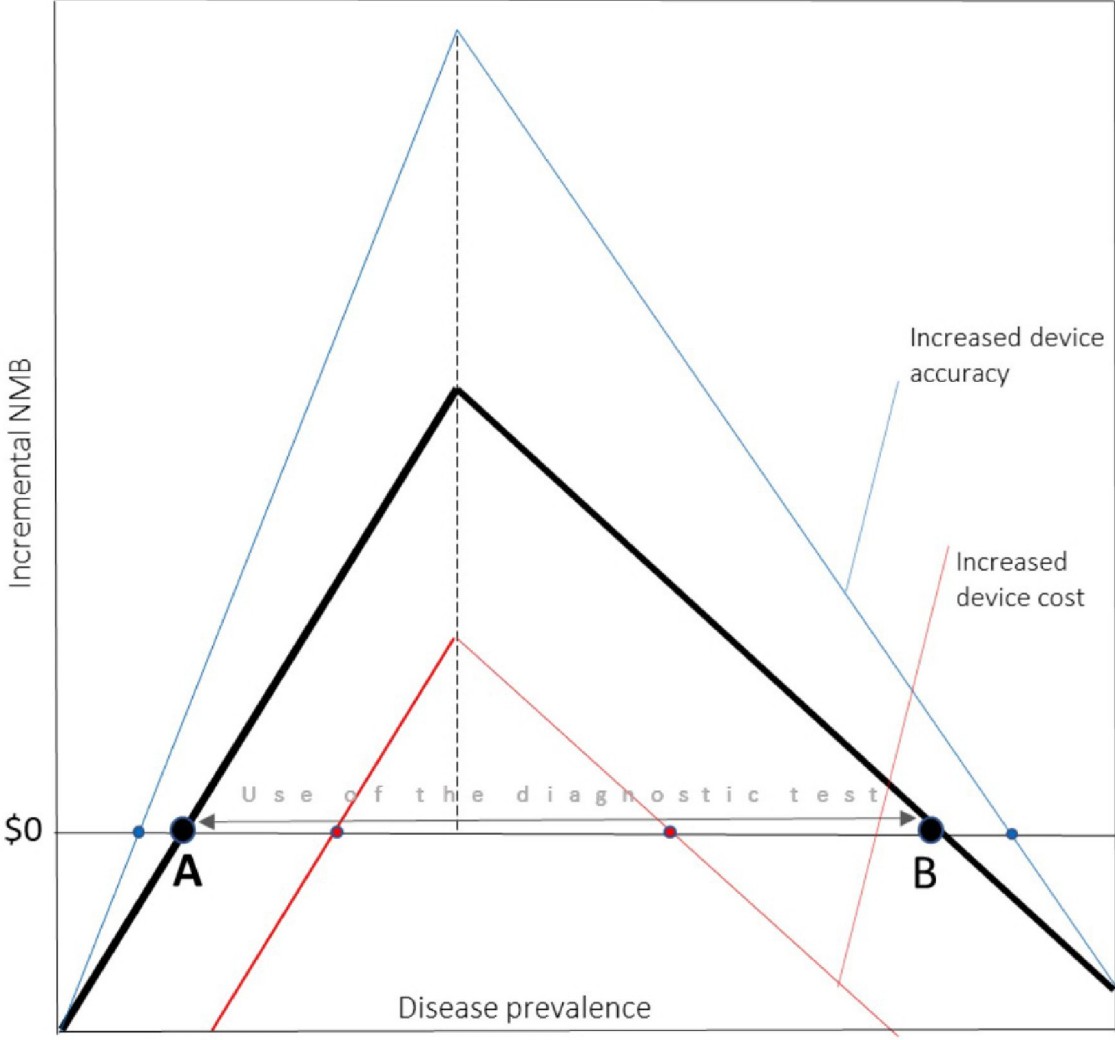

**Fig 1. Incremental net monetary benefit (NMB) of a device as a function of disease prevalence.** In Blue there is a variation of incremental net monetary benefit by an increase in diagnostic accuracy (blue). In red a variation in incremental net monetary benefit generated by an increase in cost. Source: Phelps, C.E. and A.I. Mushlin, 1988. [16].

of it use is positive with respect to the alternatives (usually, absence of treatment for prevalence lower than A, and treating all for prevalence greater than B). For prevalence lower than A or higher than B, alternative strategies would be preferred as the device has negative incremental net monetary benefit.

## Model structure

We adapted the paradigm described above for a healthcare protocol rather than for a single diagnostic device, developing a decision model to estimate the cost-effectiveness of using senti-nel screening for schistosomiasis for individuals presenting for malaria diagnosis. We built the model with a reference population of school-age children as they are the most affected by the two diseases as well as being the individuals most exposed to MDA campaigns for schistosomi-asis. Our model compared two healthcare protocols for febrile individuals arriving at health facilities: the current healthcare protocol using a diagnostic test for malaria only and MDA for Schistosomiasis vs an alternative protocol using a diagnostic test for malaria and a test for schistosomiasis.

Two decision trees were developed and combined. A decision tree for malaria diagnosis and treatment was adapted from a model by Shillcutt et al [17] where febrile patients present at a healthcare facility (Fig 2A). As determined by the sensitivity and specificity of the diagnos-tic, patients received a diagnosis of malaria or a non-malaria febrile illness (NMFI), and subse-quent treatment). We combined this with a second decision tree where a diagnostic for schistosomiasis was offered to every patient undergoing the malaria test. Based upon the results of the schistosomiasis test, patients were then additionally treated or not for schistoso-miasis (Fig 2B). Outcomes of both decision trees were summed to provide a single strategy-result.

## Comparators

We undertook three scenario analyses with variations to the health protocol to reflect practice in real contexts as well as potential cost savings resulting from, "economies of scope" derived from testing for multiple diseases simultaneously.

We summarise the three comparisons reflecting the variations to the healthcare protocol below:

1.  Comparison of a new protocol testing for malaria and a sentinel screening for schistosomia-sis (both common and currently available but as separate devices) -*named comparator 1*-vs standard protocol testing febrile patients for malaria only and using MDA (treat all or no one) for schistosomiasis.

2.  Comparison of a new protocol testing for malaria and a sentinel screening for schistosomia-sis (same features of comparator 1), but with diagnostics in one device, with consequent economies of scope (base case: 75% cost of the two separate devices)-*named comparator 2*-vs standard protocol.

3.  Comparison of a new protocol testing for malaria and a sentinel screening for schistosomia-sis using a new prototype multiplexed device with more accurate performances and diag-nostics in one device -*named comparator 3*- vs standard protocol

For comparator 1 and 2, the diagnosis for malaria was the same for both strategies and this would cancel out in an incremental evaluation. Therefore, the cost effectiveness of the model was dependent only on schistosomiasis prevalence. In contrast, for comparison 3, with improved sensitivity and specificity of the prototype for both malaria and schistosomiasis, the

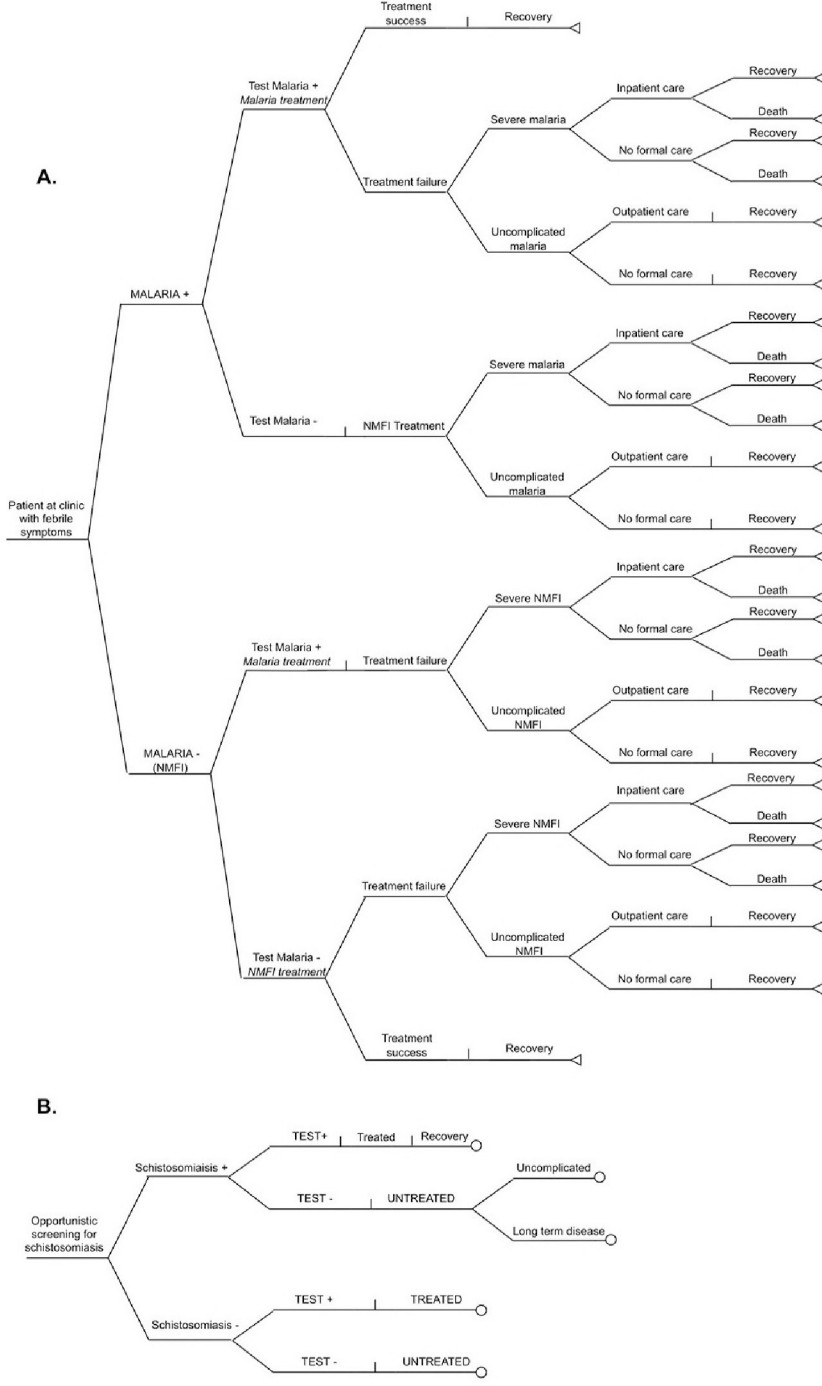

**Fig 2. Decision trees for Malaria (A) and Schistosomiasis (B).** The end point of panel B fed into decision tree A.

overall diagnoses can change, and the NMB of the device was a function of both malaria and schistosomiasis prevalence.

Usually, MDA comprises treating the whole population, regardless of their status. The decision rule to treat everyone or not is based on a prevalence threshold. For schistosomiasis, the WHO suggests a threshold of 50% prevalence in school aged children for MDA [18]. However,

as thresholds may be case-specific varying by disease and country, in the model we treated the strategies 'treat no one' and 'treat all' as boundary cases and compared this against using the test, based on cost-effectiveness.

In analysis 1, the test for schistosomiasis was undertaken either before or following the test for malaria, as it would be done with currently existing diagnostic devices, while for analysis 2 and 3 tests were taken simultaneously as in the case of the multiplex device being considered here, which embeds both assessments using the same sample (blood). The cost of the test differs between the two approaches, as one device embedding both tests will have a lower overall cost compared to two separate devices, ceteris paribus, reflecting economies of scope. If rather than a sentinel diagnosis, the screening is pursued at a different time, in a separate circumstance (e.g. by different MDA teams, as is the case in the Vector Control Division of the Uganda Ministry of Health), the overall cost would rise further.

## Model outcomes

Two different outcomes were assessed from the model to inform decision-making, namely: i) incremental cost per DALY averted, ii) incremental cost per additional correct diagnosis for the sentinel detection strategy. All outcomes were computed at different prevalence levels and presented in terms of incremental net monetary benefit (NMB) in respect to the current dominant healthcare pathway (treating everyone or no one for schistosomiasis) using a willingness to pay threshold equal to the Ugandan national GDP per capita in 2019 ($799) [19], to reflect fieldwork parameters as described in equation below.

$$\Delta NMB_{comparator_i} = NMB_{S(TEST)} - MAX(NMB_{S(TREAT\ ALL)}, NMB_{S(TREAT\ NO\ ONE)})$$

Where $\Delta NMB$ is the incremental net monetary benefit, S(TEST) is a strategy of using the test for a schistosomiasis treatment alongside a malaria test, S(TREAT NO ONE) is the strategy treating no one for schistosomiasis alongside a malaria test, and S(TRAT ALL) a strategy of treating everyone for schistosomiasis alongside a malaria test, *comparators* were described previously and consists in variations of S(TEST) alongside the malaria test.

To show how the two different diagnostic components can influence the value of the final healthcare pathway, results are presented initially only referring to schistosomiasis (analysing only panel B of Fig 2) and then, the overall healthcare pathway (schistosomiasis and malaria). When focusing solely on schistosomiasis to demonstrate the effect of economies of scope, the cost of the device containing both diagnostics halved.

## Model parameters

The sensitivity and specificity of the multiplex testing were estimated from fieldwork undertaken in Uganda between 2018 and 2020 [15,20].

Parameter estimates for malaria diagnosis, disease progression and treatment were taken from *Shillicut et al.*[17] and other recently published models (see Table 1). It was assumed that patients diagnosed positive for malaria were treated with artemisinin combination therapy (ACT) and patients negative for malaria received an antibiotic (e.g., amoxicillin). Patients who tested positive for schistosomiasis received praziquantel. This study examines costs from a payer perspective, making it relevant for public healthcare decision makers.

The cost of the device was based on personal communication with the device developers representing projections of a ready-to-market device. Most of the price of drugs was obtained from Management Sciences for Health (MSH) drug price indicator guide [21] or, when prices referring to East Africa were not available, we referred to previous studies referring to the same context (Table 1). The decomposition of a hospital first line outpatient attendances cost

**Table 1. Main Model parameters.**

| Parameter | | Source |
|---|---|---|
| Willingness to pay threshold | $799 | [19] |
| Standard Malaria RDT cost | $.9 | [28] |
| Malaria RDT prototype with sentinel screening for schistosomiasis | $1.2 | Personal communication, The University of Glasgow |
| First line care for testing initial care (outpatient cost minus test and further executive costs) | $2.00 | [22, 23,28] |
| Cost ACT therapy (full course) | $1.57 | [21,22] |
| Cost NMFI (amoxicillin full course) | $0.35 | [21] |
| Outpatient cost visit | $3.03 | [21,23] |
| Average Inpatient cost for severe malaria *general inpatient costs day\*4 days[29]* | $43.6 | [23] |
| Average Inpatient cost for severe NMFI *general inpatient costs* | $72 | [5,23] |
| Malaria RDT sensitivity | 0.98 | Personal communication, The University of Glasgow |
| Malaria RDT specificity | 0.95 | Personal communication, The University of Glasgow |
| Malaria RDT sensitivity (routinely used) | 0.87 | [30,31] |
| Malaria RDT specificity (routinely used) | 0.96 | [30,31] |
| Sensitivity sentinel screening schistosomiasis | 0.95 | Personal communication, The University of Glasgow, and literature[32] |
| Specificity sentinel screening schistosomiasis | 0.90 | Personal communication, The University of Glasgow, and literature[32] |
| Sensitivity urine test for schistosomiasis | 0.83 | [33–35] |
| Specificity urine test schistosomiasis | 0.81 | [33–35] |
| first line treatment success | 0.95 | [31] |
| probability of dying if severe and inpatient | 0.1 | [17] |
| probability of severe malaria after correct treatment | 0.1 | [17] |
| probability of dying if severe and self-treated | 0.25 | [17] |
| probability of outpatient care if uncomplicated | 0.48 | [17] |
| probability of inpatient care if severe | 0.48 | [17] |
| Disability weight moderate malaria(2 weeks event) | 0.002 | [27,36] |
| Disability weight severe malaria(2 weeks event) | 0.005 | [27,36] |
| Disability weight moderate schistosomiasis | 0.006 | [27,29] |
| Disability weight severe schistosomiasis (combined with moderate anaemia) | 0.052 | [2,27] |

in visit, drugs and diagnostic was taken from previous sources referring to Uganda [22], cost of subsequent outpatient visits corresponded to Ugandan Health centres figures [23]. Cost for subsequent inpatient or outpatient hospitalisations already included treatment costs. Given the economic costs of delivering drugs in MDA interventions [24] and the variation of such costs in the Ugandan context that can exceed twice the cost of drugs, we assumed the cost of implementing MDA higher than the cost of a 'test and treat' in line with a previous publication on Uganda [25].

All costs of treatment and healthcare system were adjusted to the 2019 price level using the corresponding annual inflation rate [19] and exchange rates following IMF for Uganda [26]. Time horizon of the analysis was one year; as malaria is an acute illness with symptoms persisting for two weeks, the disability weights were adjusted accordingly [27]. MDA coverage in base case was estimated to be 100%, values more representative to the Ugandan context were

discussed in scenario analysis. The main parameters used to generate the results of this paper are reported in Table 1. The complete list of model parameters is available in the supplementary material (S1 Table).

## Scenario analysis

The model informs policy on the value of sentinel screening for different endemic diseases in sub-Saharan Africa alongside a routine test for malaria, based upon the results of analyses simulating different scenarios to reflect contextual differences. While type of tests and potential economies of scope were already analysed in the different potential healthcare pathways, sensitivity analyses were conducted based on potentially relevant drivers for cost-effectiveness:

i.  Increase of economies of scale (decrease costing of MDA).

ii.  Diminishing the MDA coverage at 75%, reflecting minimum WHO recommendation [7]

iii.  NTD severity: the severity of the disease detected through the opportunistic test is higher (100% increase in Disability weight).

iv.  Cost-effectiveness threshold: Reduction in willingness to pay (one third).

 Scenarios i) and ii) are deemed particularly relevant. Ignoring economies of scale in rural areas (reduction in the cost per treatment of MDA -mainly related to delivery- due to the increasing the numbers treated) can deeply affect policy recommendations [37]. Even within the same country, the delivery cost can dramatically vary between different rural areas. In Uganda, it was previously assessed that the delivery costs can augment by 3 times ($ 0.19–0.69) depending on local specificities, with consequent variable weight on the final economic cost per child treated ($ 0.41–0.91, 25].

 We display economies of scale vs the cost-effectiveness of the different protocols, covering a range of MDA from $.25 (the cost of praziquantel) to $1.5. Scenario ii), by increasing the coverage could also potentially increase economies of scale, this is also treated in the discussion.

## Results

### Base case

**Schistosomiasis only perspective.** Table 2 shows the incremental net monetary benefit resulting only from schistosomiasis (referring to the marginal gain coming from the schistosomiasis part of the healthcare pathway only, Fig 2B). Regarding DALY averted, at the current level of schistosomiasis in rural Uganda (26% [38]), assuming a MDA coverage of 100% and a WTP equal to one GDP per capita in 2019 ($799), the best strategy is treating everyone, as the EVCI for the alternative protocols are all negative.

 In contrast, regarding the outcome 'correct diagnoses', testing for schistosomiasis with current diagnostics is already preferential to the current healthcare protocol of MDA for schistosomiasis (EVCI = $59.1), however, more efficient diagnostics for schistosomiasis (comparator 3) would make healthcare protocols testing for both diseases even more cost-effective (EVCI = $132.5). Looking at these healthcare pathways as a function of all ranges of schistosomiasis prevalence (Fig 3, first row), comparison 1 and 2 show that with current diagnostic technologies for schistosomiasis, the MDA strategies (treat all or no one) are always preferred. On the contrary with schistosomiasis prevalence at 5% testing with a new device with higher accuracy and economies of scope (comparator 3) would be the preferred option. For correct diagnoses, alternative healthcare protocols with current diagnostics are very similar as there is a small difference in cost relative to the effectiveness gains.

**Table 2. Net Monetary Benefit and Expected Value of Clinical Information referring on Schistosomiasis only arm, for general Schistosomiasis prevalence in rural Uganda.**

| | Total cost | DALY | NMB | EVCI |
|---|---|---|---|---|
| **DALY** | | | | |
| *Current treatment alternatives* | | | | |
| MDA (no schistosomiasis treatment) | $0.00 | 0.005 | -3.64 | |
| MDA (schistosomiasis treatment) | $0.68 | 0.000 | -0.68 | |
| *Comparison 1* | | | | |
| Test schistosomiasis | $0.99 | 0.001 | -1.61 | -0.93 |
| *Comparison 2.* | | | | |
| Test schistosomiasis (cost savings) | $0.76 | 0.001 | -1.38 | -0.70 |
| *Comparison 3.* | | | | |
| Prototype test schistosomiasis (cost savings and greater accuracy) | $0.63 | 0.000 | -0.81 | -0.13 |
| | Total cost | Correct Schistosomiasis diagnoses | NMB | EVCI |
| **Correct diagnosis schistosomiasis** | | | | |
| *Current treatment alternatives.* | | | | |
| MDA (no schistosomiasis treatment) | $0.00 | 0.74 | 591.26 | |
| MDA (schistosomiasis treatment) | $0.68 | 0.26 | 207.06 | |
| *Comparison 1.* | | | | |
| Test schistosomiasis | $0.99 | 0.82 | 650.36 | 59.10 |
| *Comparison 2.* | | | | |
| Test schistosomiasis (cost savings) | $0.76 | 0.82 | 650.58 | 59.32 |
| *Comparison 3.* | | | | |
| Prototype test schistosomiasis (Cost savings and greater accuracy) | $0.68 | 0.91 | 723.78 | 132.52 |

While testing with current technologies (comparison 1 and 2) would be preferred only at prevalence between 46% and 60%, testing with more accurate technologies is preferred for most prevalence levels.

## Overall healthcare protocol perspective

When considering the overall healthcare protocol (Table 3) at current levels of prevalence for both diseases (38% for prevalence of malaria at healthcare facilities [39] and 26% for schistosomiasis), including also malaria outcomes, the EVCI does not change significantly to replace the dominant strategy for comparator 1 and 2 (cost and DALY increased by the same amount).

By accounting for DALY averted coming from both diagnoses, the current device with economies of scope (comparator 2) would be dominant only between 4.4% and 7.1% levels of schistosomiasis prevalence (Fig 3).

In contrast, when the accuracy of the malaria device increased (comparator 3), also the EVCI raised, being positive for the referring values of malaria and schistosomiasis in Uganda ($0.65). As the number of individuals correctly diagnosed for schistosomiasis is the same, EVCI did not have any variation for this outcome. The prototype with higher accuracy becomes the dominant strategy at all levels of prevalence below 95% (Fig 3, second row).

It is relevant how, in this last case, different levels of accuracy for the prototype regarding malaria increase the overall benefits of the healthcare protocol as function of malaria, therefore all the gains are driven by malaria improvements. Fig 3, third row shows how incremental NMB is a function of both schistosomiasis and malaria prevalence, and how the value of the protocol rises with malaria prevalence.

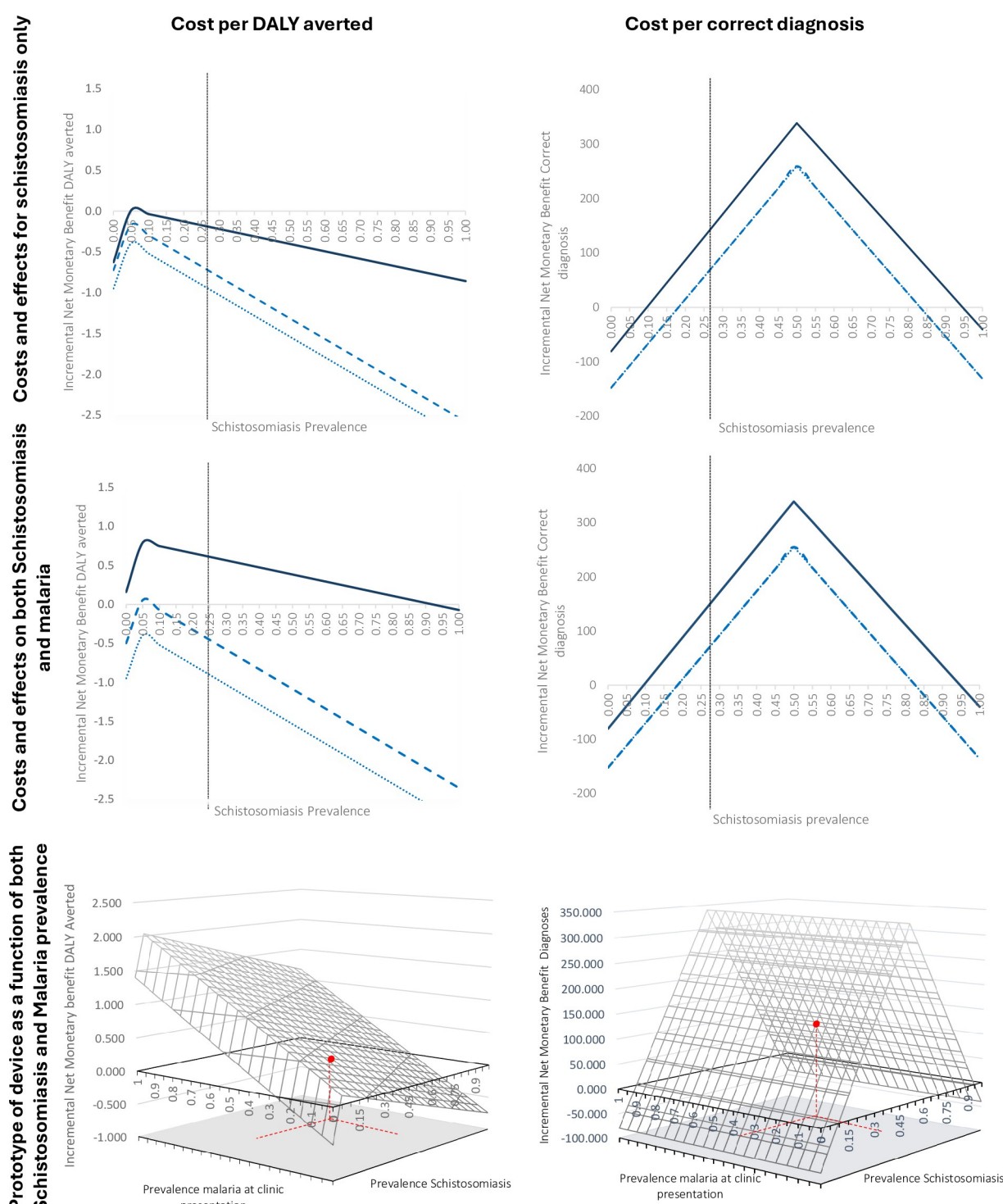

**Fig 3.** Incremental Net Monetary benefit for every comparison across the two outcomes: DALY averted (left) and Correct treatment Schistosomiasis (right). Row represents different perspectives: schistosomiasis only (A), Schistosomiasis and Malaria (B), Representation of incremental NMB of the prototype device as function of both schistosomiasis and malaria (C). Red dashed lines represent relevant prevalence levels of schistosomiasis and malaria for the rural Ugandan context. Dark blue lines represent the prototype device (comparator 3), dotted and dashed lines represent no cost savings (comparator1) and cost savings with current diagnostics (compartor2), respectively.

**Table 3.  Net Monetary Benefit and Expected Value of Clinical Information referring on both schistosomiasis and malaria, for general Schistosomiasis and prevalence of Malaria cases among febrile patients at clinics in rural Uganda.**

|  | Total cost | DALY | NMB | EVCI |
|---|---|---|---|---|
| **DALY** |  |  |  |  |
| *Current treatment alternatives* |  |  |  |  |
| MDA (no schistosomiasis treatment) | $4.77 | 0.011 | -13.76 |  |
| MDA (schistosomiasis treatment) | $5.45 | 0.007 | -10.78 |  |
| *Comparison 1* |  |  |  |  |
| Test schistosomiasis | $5.76 | 0.007 | -11.71 | -0.93 |
| *Comparison 2.* |  |  |  |  |
| Test schistosomiasis (cost savings) | $5.31 | 0.007 | -11.26 | -0.48 |
| *Comparison 3.* |  |  |  |  |
| Prototype test schistosomiasis (cost savings and greater accuracy) | $5.11 | 0.000 | -10.13 | 0.65 |
|  | Total cost | Correct Schistosomiasis diagnoses | NMB | EVCI |
| **Correct diagnosis schistosomiasis** |  |  |  |  |
| *Current treatment alternatives.* |  |  |  |  |
| MDA (no schistosomiasis treatment) | $4.77 | 0.74 | 586.5 |  |
| MDA (schistosomiasis treatment) | $5.45 | 0.26 | 202.3 |  |
| *Comparison 1.* |  |  |  |  |
| Test schistosomiasis | $5.84 | 0.82 | 645.6 | 59.1 |
| *Comparison 2.* |  |  |  |  |
| Test schistosomiasis (cost savings) | $5.31 | 0.82 | 646.0 | 59.5 |
| *Comparison 3.* |  |  |  |  |
| Prototype test schistosomiasis (Cost savings and greater accuracy) | $5.11 | 0.90 | 724.4 | 137.9 |

## Sensitivity analysis

Scenario i) representing the cost-effectiveness of different protocols across a range of economies of scale for MDA and schistosomiasis prevalence is presented in Fig 4. Regarding the outcome cost per DALY averted, a sentinel screening with current technologies (comparator 1), would be cost-effective only at high costs (meaning low economies of scale $1.25–1.5) and at low levels of prevalence (10–25%) -see blue pattern for the dominance of the comparator-. Decreasing the cost of the device (comparator 2) through economies of scope increased the range of cost-effectiveness ($1.05–1.50). When economies of scope are combined with and increase of the accuracy of the device (comparator 3), the sentinel screening is the dominant strategy for a wide range of MDA cost at low prevalences (absence of treatment is dominant only for levels of prevalence close to 0); the dominance then reduces with higher levels of prevalence. On cost per correct diagnosis, dominance depends only on schistosomiasis prevalence as difference in cost is small relative to the overall net monetary benefit. Comparator one is dominant for any level of cost between 15–80%. Comparator 2 having a small difference in costs follow the same pattern. Comparator 3 having an increase in accuracy is dominant for more range of schistosomiasis prevalence (10–90%).

Regarding other scenarios, an increase in disability weight of the disease detected opportunistically increase the slope of the right-hand side of the EVCI, reducing the level of prevalence where the new healthcare protocol is the preferred strategy (in other words, treating everyone is a dominant strategy also for lower levels of prevalence). When considering schistosomiasis only, none of these health protocols was the preferred strategy at any level of prevalence.

In contrast, when considering both diseases the new protocols with higher diagnostic performance was always preferred for levels of prevalence lower than 52%. Simulating credible

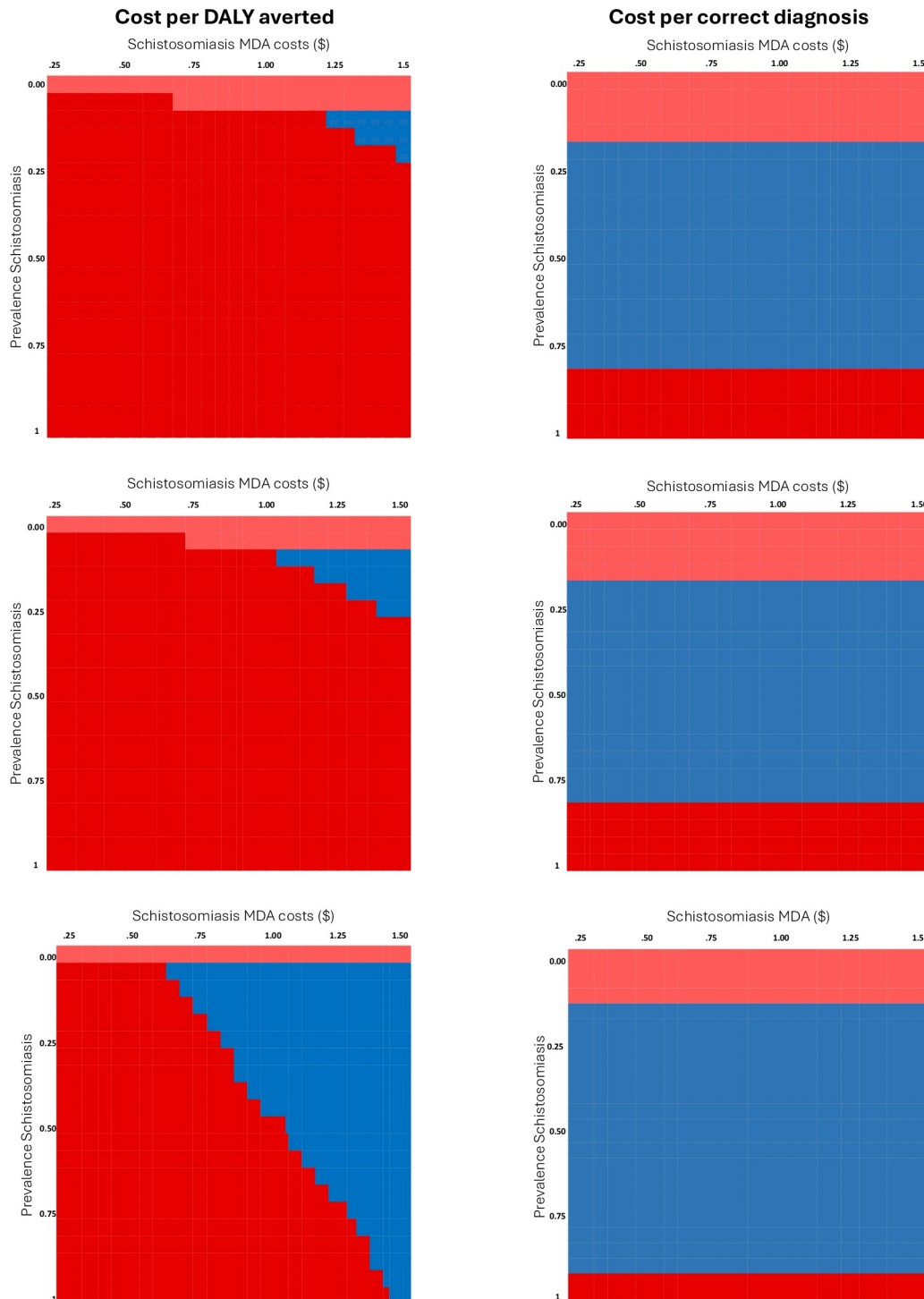

**Fig 4.** Representation of the first scenario on economies of scale for cost per DALY averted (left) and cost per correct diagnosis (right) across comparators. Colours represents different strategies: mild red (MDA-treat no one-), dark red (MDA -treat all-), blue (sentinel test).

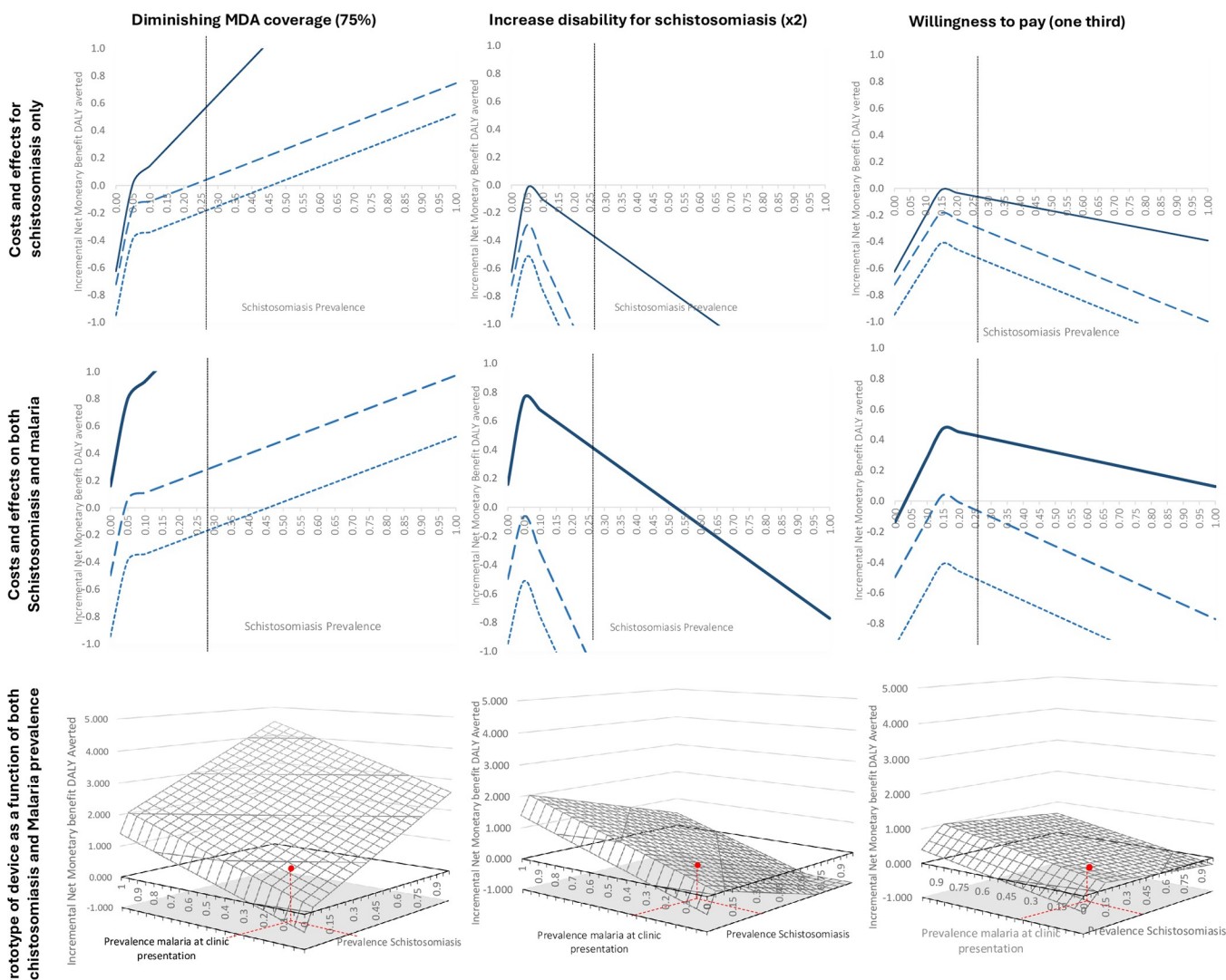

**Fig 5. Representation of the other alternative scenarios for both the outcomes.** Red dashed vertical line is in correspondence to the relevant values of prevalence for rural Uganda. Dark blue lines represent the prototype device (comparator 3), dotted and dashed lines represent no cost savings (comparator1) and cost savings with current diagnostics (compartor2), respectively.

and practical levels of MDA coverage, alternative healthcare protocols became more cost effective. Considering outcomes on schistosomiasis, sentinel screening with current technologies (comparator 1) is dominant at levels of prevalence higher than 45%. Decreasing costs and increasing accuracy (comparator 2 and 3), there are lower prevalence cut-offs.

When decreasing the willingness to pay (Fig 5, third column), the curves of all the healthcare protocols become flatter maintaining the same peak (maximum NMB). This made comparison 3 the preferable one at current levels of prevalence when considering both diseases schistosomiasis at all levels of prevalence. In contrast, when considering schistosomiasis only no protocol was preferred at current levels of prevalence when both diseases are taken into account. Scenarios representing the NMB per correct diagnosis are shown in the supplementary material (S1 Fig).

## Discussion

We present an economic model to illustrate the potential added value of "sentinel" screening policies for other endemic diseases alongside a malaria testing for febrile patients presenting at healthcare facilities. Using a case study on a prototype multiplex device able to detect two diseases at once (in this case, malaria and schistosomiasis), we found that the use of sentinel screening can be preferrable to the current treatment strategies in several settings. At estimated existing levels of prevalence in rural Uganda (38% for malaria [39] and 26% for schistosomiasis [38]) and with current commonly available diagnostics, assuming 100% coverage, MDA strategies (treat all or treat no one) remain superior when considering cost/DALY averted (the more common outcome, while cost per correct diagnosis can be used more for surveillance purposes). More accurate and low-cost diagnostics that can simultaneously be added to routine tests at patient presentation, have the potential to create alternative cost-effective healthcare protocols. Indeed, when the MDA coverage is lower than 100%, but still at recommended levels according to the WHO guidelines (e.g., 75%)[7], or MDA has low economies of scale (e.g., in remote areas) the cost-effectiveness of alternative protocols offering economies of scope is more plausible. This is relevant for the Ugandan context, where reported levels of treatment uptake during MDA were sub optimal, reaching only 45% in specific communities [40] with the main MDA challenges identified in logistics, drug shortages and community health worker attrition (all issues resolvable mainly with additional investments.

Whilst recognising MDA programmes target a broad and potentially different population than those patients that seek care, opportunistic testing (testing malaria suspects for schistosomiasis who would not otherwise get tested for schistosomiasis) could be informative: if schistosomiasis prevalence is high then MDA is superior, if between a specific prevalence range, testing wins, and if below certain prevalence ranges, the absence of treatment wins. For ease of understanding, we classified the drivers determining the prevalence cut offs for deciding diagnostic RDT vs MDA into four categories:

1) device components (accuracy -*sensitivity and specificity*- and economies of scope—*higher cost savings from detecting two diseases simultaneously increases chances of being cost effective*- of the device)

2) MDA program features (cost of treatment, economies of scale- *reduction of single treatment costs when increased the number of individuals treated*- and MDA coverage -*lower percentages make 'treat all' MDA strategy less effective than high accuracy testing in high prevalence setting*-)

3) environmental components (local capacities and expertise, disease prevalence). We first go through them below and then we discuss them in detail.

4) Perspective -*analysing the healthcare protocol overall, considering health gains coming from both diseases increase the likelihood of a new protocol including multiple detections to be cost-effective*-. However, given the higher burden of malaria, gains in EVCI from and increase in the accuracy of the device are greater for malaria. Therefore, to have an objective evaluation of a sentinel screening of the device referring to NTD only and avoid distributional gains in the schistosomiasis detection coming from malaria, we frame our discussion mainly on the 'schistosomiasis only' perspective.

The decision strategy for a different healthcare protocol rather than MDA for schistosomiasis will vary with the prevalence of the diseases affected by the protocol. In comparator 1 and 2, the malaria part of the model was the same for both pathways as well as for the baseline strategy, so the cost-effectiveness was driven only on schistosomiasis prevalence. Indeed, comparator 2 had the same features of comparator 1 but with lower 'cumulative costs', which increased the incremental monetary benefit of the comparator across all levels of schistosomiasis

prevalence (graphically shifting up the incremental NMB curves, Fig 3). Comparator 3 is the only healthcare protocol that has a full range of variations based on the prevalence of both diseases as it consisted of an increase of the performance in the detection of both diseases and going beyond a decrease in price (third row in Fig 3).

The importance of studying these effects in a variety of different settings, including different cultural settings, and both rural and urban environments cannot be under-estimated. While our three protocols presented could be perceived as three alternative meaningful scenarios, additional sensitivity analyses aimed to reflect different contexts as well as different diseases (e.g., through changes in disease severity and related disability weighting) that could be detected opportunistically. Results show that prevalence thresholds at which sentinel screening become cost-effective are context and diagnostic-specific). Furthermore, the more cost-effective strategy is dependent upon the outcome of interest. If the aim of the policy was only to correctly diagnose (and consequently treat) the population (right column Fig 3), using currently available diagnostics (comparator 2 and 3) will be cost effective for levels of prevalence between 17 and 83% of schistosomiasis, and almost always cost effective using more accurate diagnostics (comparator 3).

Integrating control programmes can provide economies of scope. However, economies of scope can also be incurred by savings in logistical (and personnel) expenses. Indeed, if rather than a sentinel diagnosis, screening takes place in on a separate occasion, for example, during surveillance programmes (e.g., by different MDA teams, as is the case in the Vector Control Division of the Uganda Ministry of Health), the overall cost would rise further for reasons associated with costs of logistic components (thus increasing the relevance of economies of scope). In this respect, at current average levels of MDA coverage and prevalence for schistosomiasis in Uganda, investments in reducing costs through economies of scope acting on device or program components (e.g., investing in multiplex devices or in other practice cost saving features) have the potential to make sentinel screening cost effective. In contrast, focusing on specific acting on MDA programme features (e.g., cost savings within certain high populated communities and consequent economies of scale), could increase gains from mass policies (scenario analysis i)).

This is important specially to evaluate and develop future policies, as without considering the potential economies of scale and economies of scope associated with interventions, it is possible that cost data will be over-generalised within economic evaluations, leading to poor policy formulation [35].

In this model, we included treatment for NMFI when a patient was tested as malaria negative and the possibility to be treated with 'informal' or 'formal' care. The value of some of these parameters have already been explored in other malaria models [5,16] but in relation to different research questions. In low-and middle-income countries, the value of sentinel screening during formal visits at medical care facilities has been studied for other morbidities and contexts, showing that it can be an effective way to increase awareness of the disease detected opportunistically [41]. This is even more important when awareness of certain specific diseases like schistosomiasis [40], increase prevention and then individual treatment uptake.

We believe that our model, combining malaria (which is among the most frequent suspected diagnosis in sub-Saharan Africa) and other diseases, can offer additional information on cost effectiveness for both surveillance and diagnostic purposes when the competing disease is endemic. In contexts where budget constraints determine how, and if, to pursue a public health surveillance, countries are more likely to prioritise surveillance on those diseases subsidised by, or of greater interest, to international donors. Notwithstanding this, such models can be used to determine whether it could be cost-effective to use sentinel screening focused on certain target populations as 'opportunistic surveillance' for less popular and more 'neglected' diseases.

The accuracy and cost of the diagnostic together with the MDA coverage and other context specific parameters should determine prevalence cut-offs to decide upon RDTs or MDA policies rather than blanket thresholds that do not consider local features.

## Further opportunities and limitations

The purpose of the study was to develop a model able to estimate the cost effectiveness of different strategies for sentinel screenings of endemic diseases alongside a test for malaria. We developed a model starting from a simple but robust design [16] analysing three health protocols executing such sentinel screening through different testing devices. This study's insights can be made adaptable to different diseases or populations through the use of the model.

Different disease combinations could be potentially more useful for diagnostic purposes. For instance, a device embedding a different NMFI alongside malaria could be beneficial for diagnostic purposes if it could rule out other competing NMFI from the bundle of suspected diagnoses, providing more accurate diagnosis and treatment. This could be readily implemented in a similar model where the decision tree in Fig 1B should be implanted within the second node of the NMFI branch in the malaria tree). In such models, when the diagnosis of specific competing NMFI is linked to the malaria prevalence, the prevalence of malaria would be a driver for policy decisions also at lower levels. To enable more comprehensive analyses on which policy makers can base their decision, results concerning additional health outcomes such as *cost per correct diagnosis from the sentinel screening* were also reported.

The model presented in this paper is highly sensitive to specific parameters (e.g., MDA coverage as well as diagnostic accuracy and costs. In this respect, there is an extensive literature stressing how context specific settings (e.g., endemicity of disease, and aim of the test -e.g., interruption of transmission or treatment of the single patient-) all have different requirements for diagnostic tests regarding schistosomiasis [42,43]. This is even more relevant in economic evaluations when key cost and prevalence differences exist within a single country [25].

In this study, we aimed to show how the cost-effectiveness of opportunistic screening is driven by main drivers such as device accuracy and environment (i.e. disease prevalence). Further, it is important to note that the sensitivity of the model to input change depends both on complexity and accuracy, and we believe that our study addresses this trade-off, based upon previous models with similar purposes assessing the cost-effectiveness of different malaria RDT [5,16].

On the relevance of economies of scale as a driver for MDA policy decisions, a recent study [44] highlighted how economies of scale can inform the cost-effectiveness of widening MDA every 6 months in communities with a low prevalence rather than targeting with MDA only specific subpopulations (e.g. school children). In addition, not only is the level of MDA coverage, but also the adherence can both influence our protocol effectiveness. The effectiveness of MDA in areas of high schistosomiasis prevalence is indeed likely to be overestimated as rural Uganda was found to have lower values than the WHO targets [40]. Parasite transmission dynamics were not modelled, but we believe that they would decrease the effectiveness of intervention; but remain not crucial for a model with a 1-year time horizon. Further, our model started from febrile patients presenting at healthcare facilities and the prevalence of malaria relates to this specific population, with other similar models starting from the same assumption [17].

Lastly, our model could also embed the additional value of avoiding treatment side effects for individuals tested negative compared to MDA. This could be modelled (Fig 2) by adding disability weights in all treated individuals. This would increase the EVCI of the test and can be particularly useful for treatment with severe side effects. However, as side effects from

praziquantel—although common—are usually mild and temporary, we opted not to model this in our schistosomiasis case study.

Finally, our sensitivity analyses are not meant to be exhaustive, but we believe they may point to the direction of the difference in results when some assumptions driving cost-effectiveness change.

## Conclusion

This paper presents a model estimating the added value of sentinel screening for an endemic disease usually treated with MDA when patients present at clinics with febrile symptoms and would be tested for malaria. We explore features influencing the cost-effectiveness of offering sentinel screening for a general endemic disease for patients receiving a test for malaria, presenting how characteristics related to the device (e.g., accuracy and economies of scope), MDA programme (e.g., economies of scale and MDA coverage), context and or perspective are the main drivers to determine the prevalence cut offs for deciding diagnostic RDT over MDA based policies. At average levels of prevalence of schistosomiasis in Uganda, a sentinel screening with current diagnostics and costs is likely to be not cost-effective. However, when we considered suboptimal levels of MDA coverage (as likely to be in the country), for high levels of schistosomiasis prevalence, economies of scope can make sentinel screenings using device with current accuracy cost-effective.

Overall, sentinel screening for neglected tropical diseases such as schistosomiasis can be a relevant cost-effective alternative to MDA, however, decisions are context and device specific. By highlighting the importance of the epidemiological setting in determining the best strategy, the model results show how blanket thresholds recommended by international guidelines may not always be the most cost-effective strategy for endemic diseases.

## Supporting information

**S1 Table. Complete list of model parameters.**
(DOCX)

**S1 Fig. Scenario analyses for the outcome cost/correct diagnoses.**
(PDF)

## Author Contributions

**Conceptualization:** Francesco Manca, Giorgio Ciminata, Eleanor Grieve, Julien Reboud, Jonathan Cooper, Emma McIntosh.

**Data curation:** Francesco Manca, Giorgio Ciminata.

**Formal analysis:** Francesco Manca, Giorgio Ciminata.

**Funding acquisition:** Julien Reboud, Jonathan Cooper, Emma McIntosh.

**Methodology:** Francesco Manca, Giorgio Ciminata, Eleanor Grieve, Julien Reboud, Jonathan Cooper, Emma McIntosh.

**Supervision:** Jonathan Cooper, Emma McIntosh.

**Writing – original draft:** Francesco Manca, Giorgio Ciminata, Eleanor Grieve, Julien Reboud, Jonathan Cooper, Emma McIntosh.

**Writing – review & editing:** Francesco Manca, Giorgio Ciminata, Eleanor Grieve, Julien Reboud, Jonathan Cooper, Emma McIntosh.

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
