## [Decision Letter · Decision Letter 0]

18 Mar 2024

Dear Dr Ciminata,

Thank you very much for submitting your manuscript "Cost-effectiveness of Screening of Endemic Diseases alongside malaria Rapid Diagnostic Testing: Implications for mass drug administration policies" for consideration at PLOS Neglected Tropical Diseases. As with all papers reviewed by the journal, your manuscript was reviewed by members of the editorial board and by several independent reviewers. The reviewers appreciated the attention to an important topic. Based on the reviews, we are likely to accept this manuscript for publication, providing that you modify the manuscript according to the review recommendations. 

Overall the comments of the reviews are favourable. The authors are suggested to respond to the concerns raised by reviewer #4 and minor concerns by other reviewers

Sincerely,

Syamal Roy

Academic Editor

Shaden Kamhawi

Editor-in-Chief

Overall the comments of the reviews are favourable. The authors are suggested to respond to the concerns raised by reviewer #4 and minor concerns by other reviewers

Reviewer's Responses to Questions

**Key Review Criteria Required for Acceptance?**

**Methods**

-Are the objectives of the study clearly articulated with a clear testable hypothesis stated?

-Is the study design appropriate to address the stated objectives?

-Is the population clearly described and appropriate for the hypothesis being tested?

-Is the sample size sufficient to ensure adequate power to address the hypothesis being tested?

-Were correct statistical analysis used to support conclusions?

-Are there concerns about ethical or regulatory requirements being met?

Reviewer #1: 1. The paper brings out a pertinent issue on combination of screening tests which in my view would greatly be cost saving and more acceptable to population and health authorities. I have worked in an environment where parallel programs for each disease created fatigue to all parties concerned! It would also strengthen the current advances in science. It could be perfected over time and that has been the case with the evolution of malaria RDT.

2. I propose consideration be made to change heading for example “cost-effectiveness of screening of endemic diseases alongside malaria diagnosis; a case of schistosomiasis”. In my understanding, it is schistosomiasis under review though a wider perspective of other endemic diseases is to limited extent mentioned (e.g helminths). Of essence, the gist of the paper is malaria and schistosomiasis. This could simultaneously change the main objective of the research

Reviewer #2: -Are the objectives of the study clearly articulated with a clear testable hypothesis stated? Yes

-Is the study design appropriate to address the stated objectives? Yes

-Is the population clearly described and appropriate for the hypothesis being tested? Yes

-Is the sample size sufficient to ensure adequate power to address the hypothesis being tested? Yes

-Were correct statistical analysis used to support conclusions? Yes

-Are there concerns about ethical or regulatory requirements being met? NO

Reviewer #3: Yes, all reviewer's comments are addressed

Reviewer #4: Review of manuscript titled “Cost-effectiveness of Screening of Endemic Diseases alongside malaria Rapid Diagnostic Testing: Implications for mass drug administration policies” 

The study aimed to present an economic model demonstrating the potential added value of "sentinel” screening for schistosomiasis, in addition to malaria diagnosis, for febrile patients at healthcare facilities. 

The authors have explored the cost-effectiveness of sentinel screening by modifying the existing healthcare protocol, considering three different comparators and three additional scenarios to generalize the outcomes for other settings and diseases. 

While I appreciate the concept behind the study involving sentinel screening, the execution and presentation are not systematically done. 

Mainly, almost all the outcomes are context and variable dependent, thereby limiting the study’s scope and reliability. 

The quality of provided images, particularly in Figure 1 and Figure 2, is suboptimal and hard to follow. Figure 3 and Figure 4 need better representation, and legends, especially for Figure 2, are lacking. 

Numerous typos, poorly formulated sentences were noted throughout the manuscript. In my opinion, the manuscript requires significant reformatting in terms of figure presentation, results, and discussion. 

Considering these concerns, I have decided to reject the manuscript in its current form.

**Results**

-Does the analysis presented match the analysis plan?

-Are the results clearly and completely presented?

-Are the figures (Tables, Images) of sufficient quality for clarity?

Reviewer #1: The decision trees do not reflect the additional benefit of screening for schistosomiasis and any other NTDs. The trees more less represent outcome of decisions around malaria illness. Iam inclined to question their relevancy to the study.

Reviewer #2: -Does the analysis presented match the analysis plan? Yes

-Are the results clearly and completely presented? yes

-Are the figures (Tables, Images) of sufficient quality for clarity? Some are not of good clarity in the downloaded version. Mentioned in the detailed review.

Reviewer #3: Yes

Reviewer #4: The analysis does not match the analysis plan

The results are not clear

Figures are also not easy to follow

**Conclusions**

-Are the conclusions supported by the data presented?

-Are the limitations of analysis clearly described?

-Do the authors discuss how these data can be helpful to advance our understanding of the topic under study?

-Is public health relevance addressed?

Reviewer #1: Iam cognizant that the paper examines costs from the payer perspective who could be government, health insurance body or the patient in case of out-pocket payment. In my view this could be okay as long as all the costs are captured.

Reviewer #2: -Are the conclusions supported by the data presented? yes

-Are the limitations of analysis clearly described? partly

-Do the authors discuss how these data can be helpful to advance our understanding of the topic under study? Yes

-Is public health relevance addressed? Yes

Reviewer #3: Yes

Reviewer #4: Conclusions are not completely supported by the data

Limitations are not highlighted

The authors do discuss how these data can help understanding the topic 

Public health relevance is discussed

**Editorial and Data Presentation Modifications?**

Reviewer #1: Nil

Reviewer #2: Have mentioned some points which if discussed could add value to the paper , Highlighted in the detailed review.

Reviewer #3: Minor changes in figures resolution.

Reviewer #4: (No Response)

**Summary and General Comments**

Reviewer #1: Nil

Reviewer #2: The manuscript PNTD-D-23-00349 entitled “ Cost-effectiveness of Screening of Endemic Diseases alongside malaria Rapid Diagnostic Testing: Implications for mass drug administration policies” is overall a well written paper. The article focuses on the additional value, of sentinel screening for schistosomiasis during routine malaria diagnosis of patients presenting to malaria clinics with fever.

The study presents the results of from comparative analytical models for the same and suggests the main points identifying the drivers enabling cost effectiveness of the approach which would allow policy deliberations regarding the cost effectiveness of the approach in different endemic regions where both schistosomiasis and malaria are endemic. The present study focuses on sub Saharan Africa where both the named diseases are co-endemic, and there remains significant morbidity and mortality especially in resource challenged settings.

The analytical model in the paper w as based on the approach of Phelps and Mushin, the highlights of which have been mentioned. The model was used to evaluate a healthcare protocol, with a reference population of school going children, as they are considered to be the most affected by these two diseases in this region and additionally most exposed to mass drug administration for schistosomiasis.

The model compared two healthcare protocols for individuals with fever 1) the prevalent health care using diagnostic test based treatment for malaria and MDA for schistosomiasis.

2) a protocol using a diagnostic test for malaria and a test for schistosomisasis.

The model was used to acess cost effectiveness of such a health protocol in the Ugandan setting via three analyses, a) Comparasion of a new protocol testing for malaria and sentinel screening for schistosomiasis with currently available devices.

b) Comparison of a new protocol testing for malaria and sentinel screening for schistosomiasis with both diagnostics in one device only.

c) Comparison of a new protocol for malaria and a sentinel screening for schistosomiasis with a new device showing higher accuracy and both diagnostics in the same device.

The analyses suggests that the accuracy and diagnostic device (as expected)along with factors like the MDA coverage and other context specific parameters should determine the prevalence cut offs for deciding diagnostic RDT or MDA based policies.

While an overall well written article the fact remains that all analyses performed will remain inextricably linked to the sensitivity, specificity and usage conditions including expertise of the operators and other environmental factors. Thus it may be difficult to compare two studies from different locations for purposes of deciding policy. Also as suggested, to extend this approach of sentinel screening to other NDTs would first require the availability of suitable devices with multi pathogen detection capability. While this is certainly desirable, it will not be easy to achieve given our incomplete information about many of these pathogens. In fact even in the case of malaria there is a huge scope for more advanced, affordable specific and sensitive devices. What attracted me about the use of sentinel screening based approaches is the avoidance of unnecessary use of drugs many of which could

 have possible side effects.

It would be nice if the authors could add a few comments based on the implications mentioned in the previous paragraph.

Minor point 

Some of the figures, such as Fig.2 appeared to be extremely fuzzy in the version downloaded. If this could be kindly checked.

Reviewer #3: Overall a good study on the feasibility of screening of endemic diseases alongside malaria Rapid

Diagnostic Testing

Reviewer #4: All together the data presented in this manuscript does not support the conclusions drawn. The study design is not fully scientifically justified.

PLOS authors have the option to publish the peer review history of their article (what does this mean?). If published, this will include your full peer review and any attached files.

Reviewer #1: Yes: Robert Basaza Kanyarutokye

Reviewer #2: No

Reviewer #3: No

Reviewer #4: No

Figure Files:

Data Requirements:

Reproducibility:

References

---

## [Editor Report · Decision Letter 1]

5 Jul 2024

Dear Dr Ciminata,

We are pleased to inform you that your manuscript 'Cost-effectiveness of sentinel screening of endemic diseases alongside malaria diagnosis: a case study in schistosomiasis' has been provisionally accepted for publication in PLOS Neglected Tropical Diseases.

Best regards,

Syamal Roy

Academic Editor

Shaden Kamhawi

Editor-in-Chief

---

## [Editor Report · Acceptance letter]

22 Jul 2024

Dear Dr Ciminata,

We are delighted to inform you that your manuscript, "Cost-effectiveness of sentinel screening of endemic diseases alongside malaria diagnosis: a case study in schistosomiasis," has been formally accepted for publication in PLOS Neglected Tropical Diseases.

Best regards,

Shaden Kamhawi

co-Editor-in-Chief

Paul Brindley

co-Editor-in-Chief
